# Comparative Transcriptome Analysis between a Resistant and a Susceptible Wild Tomato Accession in Response to *Phytophthora parasitica*

**DOI:** 10.3390/ijms19123735

**Published:** 2018-11-23

**Authors:** Zunaira Afzal Naveed, Gul Shad Ali

**Affiliations:** 1Mid-Florida Research and Education Center, University of Florida/Institute of Food and Agricultural Sciences, 2725 Binion Rd, Apopka, FL 32703, USA; zunairaafzal@ufl.edu; 2Department of Plant Pathology, University of Florida/Institute of Food and Agricultural Sciences, 2725 Binion Rd, Apopka, FL 32703, USA

**Keywords:** *Phytophthora parasitica*, *Solanum pimpinellifolium*, plant defense, plant pathogen interactions, genetic resistance, antimicrobial peptides, protease inhibitors, susceptibility factors

## Abstract

*Phytophthora parasitica* is one of the most widespread *Phytophthora* species, which is known to cause multiple diseases in tomato and is capable of infecting almost all plant parts. Our current understanding of tomato-*Phytophthora parasitica* interaction is very limited and currently nothing is known at the whole genome or transcriptome level. In this study, we have analyzed and compared the transcriptome of a resistant and a susceptible wild tomato accession in response to *P. parasitica* infection using the RNA-seq technology. We have identified 2657 and 3079 differentially expressed genes (DEGs) in treatment vs control comparison of resistant (Sp-R) and susceptible (Sp-S) samples respectively. Functional annotation of DEGs revealed substantial transcriptional reprogramming of diverse physiological and cellular processes, particularly the biotic stress responses in both Sp-R and Sp-S upon *P. parasitica* treatment. However, subtle expression differences among some core plant defense related genes were identified and their possible role in resistance development against *P. parasitica* is discussed. Our results revealed 1173 genes that were differentially expressed only in Sp-R accession upon *P. parasitica* inoculation. These exclusively found DEGs in Sp-R accession included some core plant defense genes, for example, several protease inhibitors, chitinases, defensin, PR-1, a downy mildew susceptibility factor, and so on, were all highly induced. Whereas, several *R* genes, WRKY transcriptions factors and a powdery mildew susceptibility gene (*Mlo*) were highly repressed during the resistance outcome. Analysis reported here lays out a strong foundation for future studies aimed at improving genetic resistance of tomato cultivars against to *Phytopphthora* species.

## 1. Introduction

Plants are constantly attacked by numerous microbes, insects and herbivores. Being sessile, plants can’t escape from these invaders like other motile organisms. So, they have evolved strong and sophisticated defense strategies to counter these attackers. Plants defense comprises of both constitutive preformed and induced defense responses that offers several layers of protection against pathogen invasion [1,2]. Induced defense of plants comprises of two major layers, the first one is known as pathogen associated molecular pattern (PAMP) triggered immunity (PTI) in which some molecules or structures of pathogen are perceived by plant pattern recognition receptors (PRRs), followed by an array of defense responses that combat pathogen invasion. Plant-pathogen interaction represents a very dynamic evolutionary arms race in which both plants and pathogens are rapidly co-evolving to counter each other’s survival strategies [1]. Pathogens are continuously adapting to infect their hosts, bypass host’s defense, colonize, get nutrition from host and ultimately reproduce to sustain their survival in this competitive world. To counter PTI, pathogens have developed effectors. Effectors are proteins secreted by the pathogen inside their hosts to modulate defense and facilitate pathogen invasion and infection, this intervention of plant defense is termed as effector triggered susceptibility (ETS). If the plant is immune to the attacking pathogen then the second layer of immunity called effector triggered immunity (ETI) comes in action, where the effectors themselves or the defense modulations caused by them are perceived by plants through a molecular machinery coded by the resistance genes (*R* genes) that leads to the activation of downstream components of defense ultimately leading to local cell death or hypersensitive response (HR) to prevent the spread of the pathogen.

The tomato is among the most widely cultivated crop plant that is not only consumed as raw fruit and culinary vegetable in meals but also constitutes a major agricultural industry worldwide. The United States of America is among the top tomato producing countries of the world with total annual production of 246 million cwt, valued about $1.67 billion in 2017 [3]. Apart from being an important commercial food crop, tomato is being widely used as a model plant to study different aspects of plant biology, including host-pathogen interactions. Tomato production is threatened by numerous diseases caused by a variety of pathogens that belong to all major groups of pathogens like viruses, viroids, bacteria, nematodes, fungi and oomycetes. Because of its high economic value and the fact that it could be attacked by diverse pathogens tomato pathosystem is an excellent model to study plant-pathogen interactions.

Oomycetes in the genus *Phytophthora* are responsible for a number of devastating diseases in tomatoes for example, late blight, Phytopthora root, crown and Buckeye rot caused by different *Phytophthora* species. These diseases not only damage tomato crop production but also cause major postharvest losses, threatening the tomato processing industry. *Phytophthora parasitica* is mainly known as a root and fruit pathogen of tomato associated with Phytophthora root rot and buckeye rot diseases but leaf infection, stem canker, stem girdling, collar rot, blossom blight and damping off of seedlings have also been reported in tomatoes in different parts of the world. Buckeye rot is a major tomato disease in almost all tomato growing areas of the world like India, China, many parts of Europe and USA. The disease was first reported in Florida in 1907 followed by an epidemic causing 40% loss in tomato production in an experimental field in Indiana in 1921. It is still a major problem in many tomato growing states such as Florida, Maryland and Pennsylvania. Being unique among other phytopathogens, control strategies against Phytophthora diseases are very limited. Management of Phytophthora diseases by manipulating host genetic resistance is being considered the most effective, eco-friendly and in the long run cost effective resistance development strategy. Understanding and implementation of molecular basis of tomato-*Phytophthora* interactions is very important. Identification of genetic sources of resistance against *Phytophthora* species can serve as a valuable resource for developing *Phytophthora* resistant crops in future.

Several *P. infestans* resistance genes (*Rpi*) have been identified from different *Solanum* spp., mainly wild potato species. There are five major *Rpi* genes (Ph-1–Ph-5) derived from tomatoes. Ph-1, Ph-2, Ph-3 and Ph-5 were identified from different accessions of *Solanum pimpinellifolium*, a wild tomato species whereas, the source of Ph-4 was another wild tomato species *S. habrochaites.* Efforts to develop *Phytophthora* resistant potatoes and tomatoes by using individual *Rpi* genes were not successful. Because most of these genes are very specific in recognizing their *Phytophthora* counter parts (avr genes) that are diverse not only among different *Phytophthora* spp. but also among different races and strains of same species [4,5]. Pyramiding of multiple *Rpi* genes in potato offered somewhat stable and broad-spectrum resistance against *P infestans.* Other than *R* genes, comparative transcriptome studies of *P*. *infestans* resistant *S. pimpinellifolium* enabled the identification of *P*. *infestans* resistant transcription factors SpWRKY3 [6], and a long non-coding RNA (lncRNA16397) along with glutaredoxin (SpGRX) gene [7]. Transgenic tobaccos containing another transcription factor, *SpWRKY1* were found to be resistant against *P. nicotianae* [8]. Identification of non-race specific sources of resistance and their pyramiding with multiple *R* genes can offer stable broad spectrum resistance against these rapidly evolving *Phytophthora* spp. In our laboratory, several accessions of a wild tomato species were screened against *P. parasitica* infection and some highly resistant and susceptible accessions were identified. In this study, we are comparing the transcriptome of *P. parasitica* resistant (hereafter Sp-R) and susceptible (hereafter Sp-S) accessions of *S. pimpinellifolium* to understand the molecular basis of resistance and susceptibility. Our main goal was to identity genes putatively involved in resistance against *P. parasitica.*

## 2. Results

### 2.1. P. parasitica Infection on S. Pimpinellifolium Accessions

Initial screening of thirty wild tomato accessions was done against *P. parasitica* that resulted in the identification of some resistant and susceptible accessions. We chose a highly resistant (Sp-R) and a highly susceptible (Sp-S) accession for RNA-seq analysis. Leaves from thirty-day old plants were inoculated with the *P. parasitica* zoospore suspension and symptoms were observed overtime. Localized HR-like lesion was observed on the leaves of Sp-R leading to pathogen’s growth arrest and thus no disease whereas water soaked lesions were observed on the leaves of Sp-S that resulted in severe disease development and visible pathogen growth and sporulation on the leaf surface (Figure 1). Like other *Phytophthora* spp., *P. parasitica* is a hemi-biotroph, infection studies in tomato and *A. thaliana* have revealed a short biotrophic phase (between 0 to 24 hpi) followed by quick switch to a necrotrophic phase at about 30 hpi [9,10]. To capture transcriptional changes during biotrophic and transition phases with an objective to understand molecular basis of resistance, we collected two replicates of each *P. parasitica* treated and control Sp-R and Sp-S samples at 24 hpi and 48 hpi for RNA-seq analyses.

### 2.2. Analysis of RNA-Seq Data 

126 base pair long raw reads were generated from all twelve libraries with an average of 13 million paired end reads per library; 329 million in total. After adaptor removal and quality filtering, 274 million clean reads were obtained. A very stringent criteria was used in cleaning data, all the sequences with Phred score of 27 were filtered out leaving clean reads ranging from 16 to 26 million for all samples (Table 1). Clean reads from all twelve libraries were mapped to the tomato genome SL2.50. All the samples showed overall mapping coverage of 96–97 percent. While the percentage of uniquely mapped reads of all the samples ranged between 76% to 85%. Further analysis was carried out using only the uniquely mapped reads. Detailed sequencing and mapping statistics is given in Table 1.

Genes with reads per kilo base million (RPKM) values greater than zero in both replications for each group were considered expressed. More than 20,000 genes were expressed in all the treated and control groups of both genotypes at both time points with varied number of specifically expressed genes in each group, the highest was 1048 in *P. parasitica* infected Sp-S samples at 48 hpi (Figure 2). Overall, 22,715 expressed genes were found across all 6 groups. In total, 21,789 expressed genes were present in R samples and 21,924 in Sp-S samples, among them 791 and 926 were specifically expressed across all Sp-R and Sp-S samples respectively (Figure 2).

To check the quality of our data and to have an overview of variations among all our samples, principal components analysis (PCA) was conducted using RPKM values. Principal component (PC) 1 explained 50% of the variation, PC2 comprised of 32% variance, together it explained 82% variance among 12 groups (24 samples). Samples from resistant (Sp-R) susceptible (Sp-S) accessions were separated clearly in two groups, depicting transcriptome variation between the two genotypes. *P. parasitica* treated and mock inoculated samples showed considerable variations among them, revealing the transcriptome change in response to pathogen in both Sp-R and Sp-S genotypes. Relatively less variations were observed between both treatment time points of both Sp-R and Sp-S samples, indicating some transcription reprogramming overtime in response to pathogen. PCA plots revealed very little to no variance among all sample replicates, indicating very high reproducibility of biological replicates and high-quality sequencing data (Figure 3).

### 2.3. Differential Expression Analysis

Differential expression was calculated for all treatments versus mock samples of both Sp-R and Sp-S genotypes over both time points. In addition to that, all mock and treated samples of both genotypes were compared to each other. Expressed genes with FDR *p*-value ≤ 0.001 and Log_2_ fold change >|2| were designated as differentially expressed genes (DEGs). Overall, more DEGs were identified in Sp-S treated vs. Sp-S-control compared to Sp-R-treated vs. Sp-R-control and the number of DEGs found at 24 hpi was greater than 48 hpi in all treatment vs. control as well as Sp-R vs. Sp-S comparisons. Moreover, the number of down regulated genes were greater than upregulated ones in all comparison groups (Appendix A). The highest number of DEGs, 3079 was found in Sp-S-24 hpi treated vs Sp-S-control comparison with 1132 upregulated and 1947 downregulated DEGs. Whereas, a relatively lower number of DEGs (2657) was observed in Sp-R-24 hpi treated vs. Sp-R-control comparison. In Sp-R vs. Sp-S differential expression analysis, 1158 and 889 DEGs were observed at 24 and 48 hpi respectively. We also compared controls of Sp-R and Sp-S genotypes and found 322 DEGs, the lowest count among all the comparisons. Detailed DEGs counts for all comparisons is given in Appendix A. Venn diagram analysis revealed 1173 genes were differentially expressed only in Sp-R accession upon *P. parasitica* inoculation, of them 519 and 293 DEGs were unique to 24 and 48 hpi respectively (Figure 4 and Appendix A). Among DEGs exclusively found in Sp-R accession, many important plant defense related genes were highly induced for example, Serine protease inhibitors (Solyc09g084480.2, Solyc09g084490.2 and Solyc09g089530.2) [11], Kunitz family trypsin and protease inhibitor protein (Solyc03g098790.1), Chitinases (CH) (Solyc10g074440.1, Solyc02g082930.2, Solyc05g050130.2 and Solyc02g082960.2), Arabidopsis defensin-like protein (Solyc07g009260.2) [12] and VQ motif-containing protein (Solyc02g064570.1) [13], and so on. On the other hand, many important defense gene like, *R* genes, calcium signaling genes, BAK1-interacting receptor and so on, were found downregulated. *Mlo* that is a well-known powdery mildew susceptibility [14] gene (Solyc03g095650.2) was found highly suppressed in Sp-R (Appendix A).

Heat map based on hierarchical cluster analysis of expressed genes across all samples, revealed specific expression trends of some core defense genes across Sp-R vs. Sp-S accessions as well as control vs. infected samples of both accessions. Among the top 50 highly correlated genes across samples, 22 clustered together to reveal higher expression levels in all four mock samples compared to very low expression in all eight *P. parasitica* treated samples regardless of genotype and timepoints (Figure 5). Among rest of the clusters, several core plant defense related genes were seen highly induced in both resistant and susceptible accessions upon *P. parasitica* treatment.

In *P. parasitica* treated samples, 18 genes were found highly upregulated in Sp-S and comparatively less induced in Sp-R at both 24 and 48 hpi. 14 out of these 18 were very important plant defense regulators including 5 pathogenesis related proteins (Solyc09g007010.1, Solyc04g064880.2, Solyc01g106620.2, Solyc01g097270.2 and Solyc01g097240.2), two chitinases CHI3 (Solyc02g082920.2), CHI-B (Solyc10g055800.1), LOX1 (Solyc08g029000.2), MLP423 (Solyc09g090980.2) and BG1 (Solyc10g079860.1) [15,16,17,18]. Another important plant defense gene, rubisco activase (Solyc10g086580.1) was expressed in both Sp-R and Sp-S *P. parasitica* treated plants only at 48 hpi [19]. Two genes, 2-oxoglutarate (2OG) and Fe(II)-dependent oxygenase superfamily protein (Solyc10g086580.1.1) and GTP binding elongation factor (Solyc11g069700.1) were found highly induced only in *P. parasitica* infected Sp-R samples at both time points. Few genes were induced in all treatments but induction was much higher in Sp-R plants compared to Sp-S. For example, a heat shock protein HSP70 (Solyc09g010630.2) and PHE ammonia lyase 1 or PAL1 (Solyc09g007920.2) were most highly upregulated in Sp-R at 24 hpi. Highest induction of Peroxidase 2 (Solyc01g006300.2) was found in Sp-R at 48 hpi (Figure 5). All these genes are well known to be involved in plant defense against oomycete pathogens particularly *Phytophthora* spp. [20,21,22].

### 2.4. GO Enrichment Analysis of DEGs

Gene Ontology (GO) term enrichment analysis of 3208 Sp-R and 3426 Sp-S DEGs by Singular Enrichment Analysis (SEA) tool using *S. lycopersicum* ITAG2.4 as background reference annotated 2301 and 2485 DEGs respectively. Significant enrichment of 22 and 30 GO terms was revealed among Sp-R and Sp-S DEGs respectively. The most significantly enriched GO term, common among DEGs of both Sp-R and Sp-S genotype, was catalytic activity. Other significantly abundant molecular function (MF) terms were oxidoreductase activity, transferase activity, oxygen binding and hydrolase activity (Table 2). Biological processes (BP) terms were response to stimulus, cellular and metabolic processes, biological regulation and regulation of biological processes, and cellular component (CC) terms like cell, cell part organelle and macromolecular complex were present among DEGs of both Sp-R and Sp-S genotypes, but they were not significantly enriched (Appendix A).

Comparisons of Sp-R, Sp-S and Sp-R vs. Sp-S SEA results revealed some interesting trends. For example, peptidase and endopeptidase inhibitor activity terms were only present in Sp-R and glutathione transferase activity was specific to Sp-S. Significant enrichment of ten different transport activity related GO terms specifically assigned to Sp-S DEGs. Whereas, enrichment of carboxylesterase, cellulose synthase activity, peptidase and endopeptidase inhibitor activity were unique to be differentially expressed in Sp-R genotype only (Table 2).

### 2.5. Parametric Gene Set Enrichment Analysis Based on GO

To gain more insight about functional annotations of DEGs with relevance to their expression levels across all the groups at each time point separately, parametric analysis of gene set enrichment (PAGE) was done for all six (*P. parasitica* treated vs. mock samples of both genotypes and *P. parasitica* treated Sp-R vs Sp-S samples at both time points) comparisons. PAGE is an efficient and sensitive method that ranks annotated gene clusters according to their expression levels.

Out of total 6101 wild tomato DEGs found across all six comparisons of *P. parasitica* treated samples, 339 unique GO terms were assigned to 3654 DEGs. PAGE analysis revealed, 62 and 51 significant GO terms among Sp-S DEGs, 23 and 19 in Sp-R DEGs and 14 and 26 in Sp-R vs Sp-S DEGs at 24 hpi and 48 hpi respectively. Greater number of significant GO terms were found among Sp-S pathogen treated vs. control groups that indicates more intense transcriptional reprogramming in susceptible genotypes upon infection with *P. parasitica.*

Most of the terms related to regular physiological activity, like cellular processes, cellular component organization, localization, transporter activity, cell and cell parts, were found to be downregulated among all groups, with few exceptions in Sp-R vs. Sp-S groups. Binding, cell wall organization and or biogenesis, catalytic activity, enzyme regulator activity and transcription regulator activity were found to be upregulated in most of the groups. Metabolic processes were downregulated in Sp-R, whereas they were upregulated in Sp-S genotypes (Figure 6A).

DEGs of all six groups were found to be enriched in some important plant defense related GO categories with varied expression trends. Antioxidant activity was found highly upregulated in both Sp-R and Sp-S samples in response to *P. parasitica* infection (Figure 6A). Cell death was found to be progressively upregulated from 24 to 48 hpi in Sp-S and was downregulated in Sp-R at 24 hpi but highly up at 48 hpi. Response to stimulus was found to be upregulated across all comparisons in the bar graph (Figure 6A).

Cross comparison of GO biological processes hierarchy between Sp-R and Sp-S revealed induction of genes related to response to stimulus in both Sp-R and Sp-S accessions upon *P. parasitica* infection but there were significant differences in the downstream hierarchy. Response to biotic stimulus was upregulated in both resistant and susceptible accessions but response to wounding and response to external stimulus were specifically only in resistant accession (Figure 6B,C). Response to wounding comprised of 11 wound induced serine-type endopeptidase inhibitors (SIPs), whereas other plant defense genes like defensin, hydrolyses, kinases and *R* genes were enriching the response to stress and external stimulus categories (Appendix A). Differential expression of many different SIP genes was seen in control vs. infected samples of both Sp-R and Sp-S and in Sp-R vs. Sp-S comparisons but the number of induced SIPs were more in Sp-R as compared to Sp-S. Out of total ten highly induced SIPs in response to *P. parasitica* treatment, six were significantly upregulated in only Sp-R, two were specific to Sp-S and two were induced both in Sp-R and Sp-S (Figure 7). Defensin (Solyc07g009260.2), another important defense related gene was only induced in Sp-R at 24 hpi. An NB-ARC domain containing *R* gene (Solyc06g008790.2) was constitutively expressed at low level (RPKM around 2) in only Sp-R in both control and treated samples.

Overall, oxidoreductase activity was found significantly enriched among DEGs of both Sp-R and Sp-S in response to *P. parasitica* (Table 2). Further dissection revealed significant induction of genes categorized in daughter GO term oxidoreductase activity, acting on peroxide as acceptor and peroxidase activity were induced in both Sp-R and Sp-S whereas oxidoreductase activity, acting on the CH-CH group of donors, NAD or NADP as acceptor was found downregulated only in Sp-S (Appendix A). Significant differential induction of these core defense components in Sp-R compared to Sp-S make them potential candidates responsible for resistance against *P. parasitica* in Sp-R accession.

### 2.6. KEGG Annotations

To visualize the functional involvement of DEGs in biological pathways and to look for the differences between successful plant defense and disease development responses of Sp-R and Sp-S wild tomato accessions in response to *P. parasitica* infection at 24 and 48 hpi, Kyoto Encyclopedia of Genes and Genomes (KEGG) ontology of DEGs was done by using KEGG automatic annotation server (KAAS) and assigned KO terms for each individual group, were mapped on KEGG pathways using Pathview. Most of the DEGs showed common trend of up or down regulation in both Sp-R and Sp-S at each time point in general physiological pathways as well as plant defense related pathways with few exceptions.

Overall, 30 spots were mapped on the mitogen activated protein kinase (MAPK) signaling pathway (Figure 8). All mapped genes in ethylene responsive defense response were found upregulated in both Sp-R and Sp-S, except MAPK 3/6 that was downregulated in R and was not found among S DEGs. Some core plant defense related components, like PR1, ERF1, EBF1/2, PP2C, ETR/ERS were found upregulated, whereas, SnRK2 and CAM4 were commonly downregulated in both Sp-R and Sp-S accessions at both time points. Contrasting trends of up and down regulation between Sp-R and Sp-S were shown by only three genes. FLS2 that was found upregulated in Sp-S at both time points was downregulated in Sp-R at only 24 hpi, CAT1 that is a component of stress tolerant response was down regulated both in Sp-R and Sp-S at 24 hpi but was found upregulated only in Sp-S at 48 hpi. Upstream to CAT1, abscisic acid responsive PYR/PYL was up regulated in Sp-S at both time points but in Sp-R it showed contrasting trend of up regulation at 24 hpi downregulation at 48 hpi. WRKY2229 and WRKY33 were found downregulated only in Sp-R and wasn’t differentially expressed in Sp-S. MPK1/2/7/14, ChiB, RTE1 and ACS6 were found up regulated in both Sp-R and Sp-S at 48 hpi but were not found in Sp-S at 24 hpi. Cell death related OX1 was found upregulated only in Sp-R at 24 hpi and RbohD was only found upregulated at both time points in Sp-S (Figure 8).

Most of the plant hormone signal transduction pathway was represented by DEGs in both genotype with only few differences. Auxin responsive AUX1 was found upregulated in Sp-R and down regulated in Sp-S at both time points. Contrastingly, GH3 and JAZ were down in Sp-R and up in Sp-S. NPR1 was downregulated in both Sp-R and Sp-S whereas its regulator TGA transcription factor was found up only in Sp-S at 48 hpi. Downstream TGA there are PR1 genes possibly responsible for disease resistance were up in both Sp-R and Sp-S at both time points (Figure 9).

### 2.7. Visualization of DEGs in Plant Pathways

To visualize the functional involvement of DEGs in plant biotic stress specific pathways and to look for differences between Sp-R and Sp-S genotypes upon *P. parasitica* infection, MapMan pathway analysis tool was used [23]. In total, 982 and 970 DEGs of Sp-R and Sp-S accessions were mapped on whole plant biotic stresses related processes map respectively, depicting modulation different categories of plant defense in both accessions upon *P. parasitica* infection. Overall, no major differences were seen between Sp-R and Sp-S except a clear difference in signaling related gene that were more induced in Sp-S as compared to Sp-R. Upon focusing on signaling pathways it was revealed that the difference in expression trends were coming from receptor like kinases particularly LRRs and DUF26 domain containing kinases that are mostly upregulated in Sp-S and downregulated in Sp-R (Appendix A). DUF26 is a big family of secreted proteins having a plant specific cysteine rich motif that has been reported to be involved in antifungal activity [24]. To visualize the expression of all DUF26 coding DEGs, a hierarchal clustering heat map based on expression profiles was generated that showed most Sp-S DUF26 genes to be more upregulated at 24 hpi and their upregulation was comparatively less at 48 hpi. Some of them were seen downregulated in all *P. parasitica* treated samples of both genotypes. Four DUF26 genes (Solyc07g063710.1, Solyc07g063750.2, Solyc040796690.2 and Solyc07g063750.2) showed upregulation in Sp-R accessions and downregulation in Sp-S at both time points (Figure 10), these might be specifically related to the resistance response in Sp-R accession against *P. parasitica*.

### 2.8. Validation of RNA-Seq Results with QRTPCR

To verify differential expression analysis based on RNA-seq data, transcriptional levels of selected DEGs that were potentially involved in defense against *P. parasitica* and represent different expression profiles across *P. parasitica* treated and control samples of both Sp-R and Sp-S, were determined by qRT-PCR analysis. For example, 2-oxoglutarate (2OG) and Fe(II)-dependent oxygenase (Solyc07g043420.2), lipoxygenase 1 (Solyc08g029000.2), heat shock protein 70 (Solyc09g010630.2), elongation factor (Solyc11g069700.1) and papain family cysteine protease (Solyc04g080960.2). Relative expression values of all the tested genes were calculated by using constitutively expressed Actin gene. Expression profiles of all tested genes resulted from qRT-PCR agreed with our RNA-seq data analysis results (Figure 11).

## 3. Discussion

*Solanum pimpinellifolium* is a wild tomato species that is considered among the closest relatives of the cultivated tomato. Different accessions of *S. pimpinellifolium* have been reported to carry resistance against several diseases that threaten the production of cultivated tomato [7]. In this study we have investigated transcriptional reprogramming triggered by *P. parasitica* in resistant (Sp-R) and susceptible (Sp-S) *S. pimpinellifolium* accessions with an aim to identify genes putatively involved in resistance development in Sp-R against *P. parasitica.*

Differential expression analysis was done on treatments vs. mock samples of both Sp-R and Sp-S genotypes over both time points, to identify the specific expression changes brought out by *P. parasitica* in both accessions during biotic and switch to necrotrophic phases of infection. Major transcriptional reprogramming of plant defense related mechanisms as well as diverse cellular and metabolic processes was observed in both Sp-S and Sp-R in response to *P. parasitica* infection (Figure 6, Table 2 and Appendix A). Number of identified DEGs in treatment vs. control comparisons were higher in Sp-S so does the number of assigned GO terms (Appendix A), depicting more intense transcriptional reprogramming during susceptible interaction compared to resistance development against *P. parasitica*. The mechanism of plant’s response to pathogens is a complex phenomenon that involves interconnected network of changes in several regular physiological processes [25] that is clearly reflected from the transcriptome based studies of several diverse plant-pathogen systems [26,27,28,29,30]. Regarding plant defense, we found differential expression of several genes involved in multiple plant immunity related mechanisms in both Sp-R and Sp-S. However, our focus was to identify induced genes specifically related to development of resistance response in Sp-R against *Phytophthora parasitica*.

### 3.1. Antimicrobial Genes in Relevance to Resistance Response against P. parasitica 

The class one PR genes are well known for their involvement in plant defense against multiple pathogens. Overexpression of PR-1a gene in transgenic tobacco plants resulted in enhanced resistance against *P. parasitica* [31] whereas, transgenic tobacco plants with silenced PR-1 genes were found to be highly susceptible to *P. parasitica* as compared to the controls plants [32]. Thus, it could be speculated that PR-1 act as a positive regulator of plant resistance against *P. parasitica.* PR-1 is also considered as SA signature and its enhanced expression is linked to systemic acquired resistance (SAR) [33]. In the present study, we found induction of a PR-1 gene only during the resistance response against *P. parasitica* in Sp-R accession (Figure 7).

Other PR genes encoding antimicrobial compounds, for example, proteases, chitinases and proteinase inhibitors, constitute the basal defense in plants. Antimicrobial activities essentially serve as a first line of host’s defense against the invading pathogens. Many of them are highly induced upon perception of pathogen’s elicitors or MAMPs thus, could be considered as a component of PTI [34].

Chitinases are considered to be antifungal compounds because they have been reported to degrade chitin in the fungal cell walls, but they have also been found effective against oomycetes [35,36]. Four different chitinases were found highly induced in Sp-R upon *P. parasitica* treatment (Figure 7). A chitinase gene (*ChiIV3*) in Capsicum annuum was found to be induced by *Phytophthora capsici* was shown to have an inhibitory effect on the growth of *P. capsica*. Moreover, ChiIV3 was also found to act as a chitin (unknown) receptor that trigger cell death and defense signaling upon perceiving *P. capsica* attack [35]. In addition to plants, chitinases from a biocontrol organism from *Trichoderma* spp. have also been found effective against *P. parasitica* [36]. Enhanced induction of chitinases specifically during resistance outcome against *P. parasitica* make them potential anti-phytophthora candidates.

Protease inhibitors have been proposed to protect the plant against pathogen’s proteins and proteases [11]. Antimicrobial activity of plant proteinase inhibitors was first observed in tomato-*Phytophtora* interaction where induced trypsin and chymotrypsin proteinase inhibitors were found to corelate with resistance response of plant against *P. infestans* [11]. PAGE analysis revealed significant induction of biotic stress related genes in both Sp-R and Sp-S but response to wounding, stress and external stimulus were significantly induced only in Sp-R during resistance outcome against *P. parasitica* (Figure 6, Appendix A). Further dissection of these exclusively enriched GO terms revealed significant upregulation of antimicrobial activity related genes like wound-induced SIPs, Kunitz family protease inhibitor protein and defensin (Figure 7). Four and six different SIPs homologues were specifically induced in Sp-S and Sp-R tomatoes during susceptible and resistant interaction with *P. parasitica*. Significant induction of SIPs was observed in potato in response to *P. infestans* and wounding.

Plant defensins is another important subfamily of antimicrobial proteins. Defensins are a family of small diverse antimicrobial peptides ubiquitous to throughout the plant kingdom. They are well known for their role in plant defense against a wide variety of pathogens including different *Phytophthora* species. Transgenic tomato expressing a chili defensin gene showed increased resistance against *P. infestans* [37]. A wild tobacco defensin (NmDef02) presented very strong antimicrobial activity against *P. parasitica.* Transgenic tobacco and potato plants expressing this *NmDef02* gene showed enhanced resistance to *P. parasitica.* and *P. infestans* [38]. In our study six SIPs and one defensin were specifically induced in Sp-R accession during resistant response against *P. parasitica*. Significant induction of SIPs and defensin genes specifically in Sp-R make them potential candidates that might have some essential contribution in resistance development by serving as antimicrobial agents against *P. parasitica*.

### 3.2. Contrasting Expression of Downy Mildew Susceptibility Factor in Response to P. parasitica

Another gene (Solyc10g086580.1) annotated as 2-oxoglutarate (2OG) and Fe(II)-dependent oxygenase protein presented a contrasting trend; significant induction in Sp-R and repression in Sp-S during resistant and susceptible responses to *P. parasitica* infection respectively (Figure 4 and Figure 7). In Arabidopsis, a downy mildew resistance gene (DMR6) encoding a 2-oxoglutarate (2OG) and Fe(II)-dependent oxygenase and other DMR6-like oxygenases (DLO1 and DLO2) are reported to be associated with plant defense but these are required for susceptibility of Arabidopsis to downy mildew pathogen *Hyaloperonospora parasitica.* Arabidopsis mutants lacking DMR6 showed enhanced resistance to *Phytophthora capsica*. Enhanced expression of DMR6 was observed both in case of compatible and incompatible Arabidopsis-*H. parasitica* interactions with more strong activation during early stages of incompatible interaction compared to compatible interaction [39]. Induced expression of an immune suppressor of related oomycete pathogens leading to resistance response to *P. parasitca* could be attributed to enhanced defense leading to pathogen arrest in Sp-R, whereas decreased expression in case of susceptible response could be either a counter defense strategy executed by *P. parasitca* or repression of a susceptibility factor by the plant to execute defense. Whether the tomato DMR6 identified here is a susceptibility factor for *P. parasitca* or not, cannot be concluded solely on the basis expression studies.

### 3.3. Induction of Plant Heat Stress Tolerance Related Genes

A gene encoding GTP binding elongation factor Tu (Solyc11g069700.1) was found highly induced only in Sp-R during resistance development against *P. parasitica*. These kind of elongation factors serve as translation factors in protein biosynthesis and have been reported to play an essential role in plant heat stress tolerance [40] but their role in plant defense response is unknown. Another component of plant heat stress tolerance, HSP70 a heat shock protein (Solyc09g010630.2) was found significantly upregulated in both Sp-S and Sp-R but highest induction was seen in Sp-R at 24 hpi. HSP70 is reported to play an essential role in plant defense by interacting with MAPKs and is required for the induction of INF1 (a *P. infestans* elicitor) mediated cell death [20]. Recently, in planta association between HSP70 and a *P. infestans* effector Pi23226 was shown to induced cell death in *Nicotiana benthamiana* [41].

### 3.4. R Genes

Sp-R accession presented HR like response against *P. parasitica* infection that is generally considered an event of ETI activated by recognition of specific pathogen effector by particular *R* gene. Interestingly most of the *R* genes were either found to be downregulated or very sparsely expressed in Sp-R upon treatment with *P. parasitica*. On the contrary many *R* genes were found highly upregulated in the susceptible accession Sp-S upon *P. parasitica* infection (Appendix A). A non-traditional *R* gene called *Mlo*, was found highly suppressed only in Sp-R at 24 hpi and slightly induced at 48 hpi as well as during susceptible interaction of Sp-S to *P. parasitica* (Figure 7). *Mlo* is widely known as susceptibility factor for powdery mildew disease. Loss of function of a tomato *Mlo* resulted in enhanced resistance against powdery mildew [14]. From our data, it could be speculated that resistance outcome in Sp-R could be the result of recessive resistance instead of dominant *R* gene mediated resistance [42].

Transcriptional reprogramming of many cellular and physiological processes and all components of biotic stress responses especially high induction of several defense related genes, suppression of susceptibility factors, modulation of phytohormone signaling, proteolysis and signaling in Sp-R accession indicates significant activation of multiple defense mechanisms leading to resistant outcome against *P. parasitica*. Functional significance of defense related genes identified here needs further investigation.

## 4. Materials and Methods

### 4.1. P. parasitica Inoculation, RNA Extraction and Sequencing

A highly infectious *P. parasitica* strain (12-1) was grown on solid V8 medium plates for 10 days. Zoospores were induced by applying dehydration stress on cultures from day 5 to day 7. On the 10th day cultured plates were flooded with sterilized water to induce release of zoospores and diluted up to 10^5^ cells/mL. *S. pimpinellifolium* accessions were obtained from UC DAVIS C.M. Rick Tomato Genetics Resource Center. After initial screening of these accessions against *P. parasitica,* a highly resistant accession LA2093 (Sp-R) and a highly susceptible accession LA1581 (Sp-S) were used for further studies. Leaves of thirty days old seedlings of tomatoes grown under controlled conditions in sterilized soil were inoculated with zoospores. Two spots 10 µl of zoospore suspension were placed on the underside of the leaf (each side of midrib) with the help of pipettor. Mock inoculation was done by using sterilized water without zoospores. Around two hundred spots were inoculated in each accession and experiment was repeated three times. After three days post inoculation (72 hpi), a spot was counted as HR if the lesion was confined to the inoculation spot, whereas a spot was considered a diseased spot if a water-soaked lesion was observed on the area at least five times bigger than the initial inoculation site. For RNA extraction, leaf samples from treated and untreated plants were collected at 24 hpi and 48 hpi. Leaves were thoroughly washed, immediately frozen in liquid nitrogen and stored in −80 ^°^C until further use. Total RNA was extracted from frozen leaf tissues using a QIAGEN RNeasy kit (QIAGEN, Valencia, CA, USA). Twelve cDNA libraries were constructed and sequenced using Illumina Hiseq 2500 platform (Illumina Inc., San Diego, CA, USA).

### 4.2. RNA-Seq Analysis

Most of the sequencing data analysis was done by using CLC Genomics Workbench program (Version 9.5.3, QIAGEN, Copenhagen, Denmark). Raw data files were imported as paired end data for each individual sample. Quality of raw data was checked by generating quality check (QC) reports for each sample. Adapters were removed, and sequences were further trimmed and filtered using quality trim cutoff of 0.005 (equivalent to Phred score 27) and minimum read length of 20 bp. All sequence libraries were rechecked by generating QC reports to make sure that trimming and quality filters were stringent enough. Clean reads from all samples were mapped to *S. lycopersicum* genome (SL2.50) with mismatch cost of 2 and insertion cost of 3. Genes with RPKM values greater than zero were considered expressed. Differential expression analysis was done using the RNA-Seq tool of a CLC genomics workbench that is based on generalized linear model assuming that read counts follow a negative binomial distribution. This method is similar to what is used in DESeq and edgeR packages and by far the best reported analysis for differential expression [43]. A gene was considered differentially expressed if its FDR *p*-value ≤ 0.001 and Log_2_ fold change >|2|. Heat map was based on normalized log CPM values (CLC Genomics Manual) using Euclidean distance and complete linkage of clusters, 50 fixed number of features and minimum counts of 10 in at least one sample. All Venn Diagram analyses were done by using the webtool (http://bioinformatics.psb.ugent.be/webtools/Venn/). ITAG annotations were extracted for all differentially expressed genes (DEGs). Gene Ontology (GO) term enrichment analysis and Parametric analysis of gene set enrichment (PAGE) was done by feeding ITAG IDs to the Agriculture gene ontology (AgriGO) tool (http://bioinfo.cau.edu.cn/agriGO). AgriGO is an integrated online tool designed for GO term enrichment analysis of agricultural species [44]. Cross comparisons of individual SEA results of Sp-R and Sp-S DEGs was done by using SEACOMPARE (Cross comparison of SEA) tool of AgriGO.

In addition to ITAG annotations, Kyoto Encyclopedia of Genes and Genomes (KEGG) terms for DEGs were obtained by feeding DEGs sequences to KAAS (KEGG automatic annotation server) to assign KO terms [45] using the Nightshade family (sly, nta, ath, aly) database options. Pathview was used to generate KEGG pathways [46]. Mapman software was utilized to view DEGs in biological plant specific processes [23,47].

### 4.3. Validation of RNA-Seq Results

To validate differential expression results obtained by using RNA-seq data, top DEGs that were potentially involved in the development of resistance response in R against *P. parasitica* were subjected to quantitative real time PCR by using a previously published method [48,49]. Gene specific primer pairs (Appendix A) were designed using NCBI primer BLAST tool (https://www.ncbi.nlm.nih.gov/tools/primer-blast/). Actin gene was used as a reference to calculate the relative expression values. To test the statistically significant differential expression ANOVA followed by post hoc *t* tests (Bonferroni correction applied) were performed separately among Sp-R and Sp-S samples. Melt curve analysis was done to verify specific amplification of a single gene product by each primer pair.

## Figures and Tables

**Figure 1 ijms-19-03735-f001:**
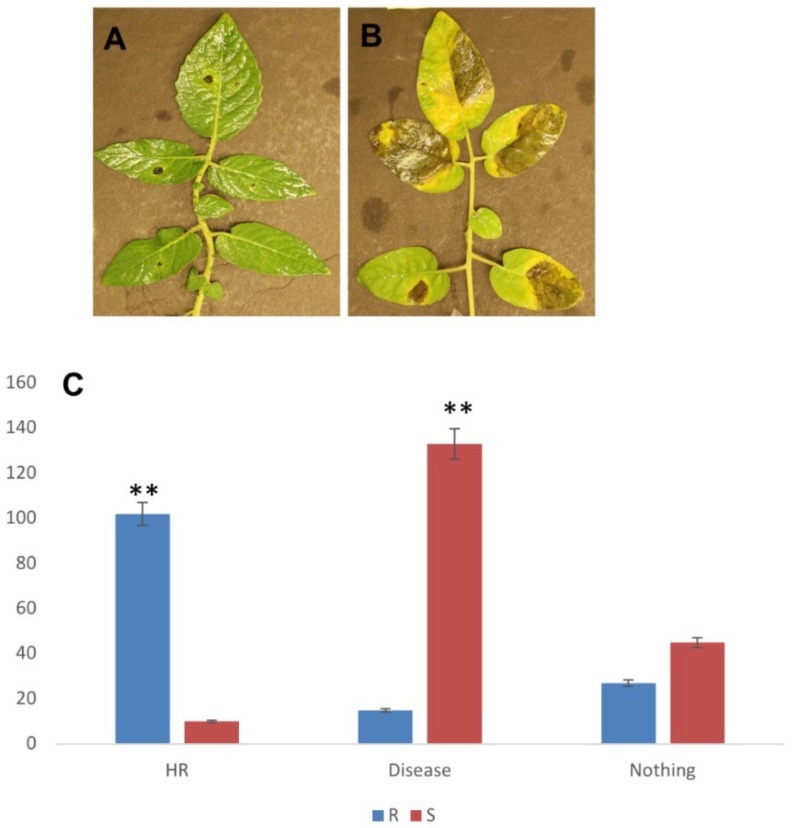
**(A)** Hypersensitive response (HR) on the leaves of *S. pimpinellifolium* accession (Sp-R) resistant to *P. parasitica* and **(B)** Disease symptoms on *S. pimpinellifolium* susceptible (Sp-S) accession in response to *P. parasitica.*
**(C)** Disease and HR spots on Sp-S and Sp-R leaves after 72 h of *P. parasitica* inoculation. ** indicates *p* < 0.05.

**Figure 2 ijms-19-03735-f002:**
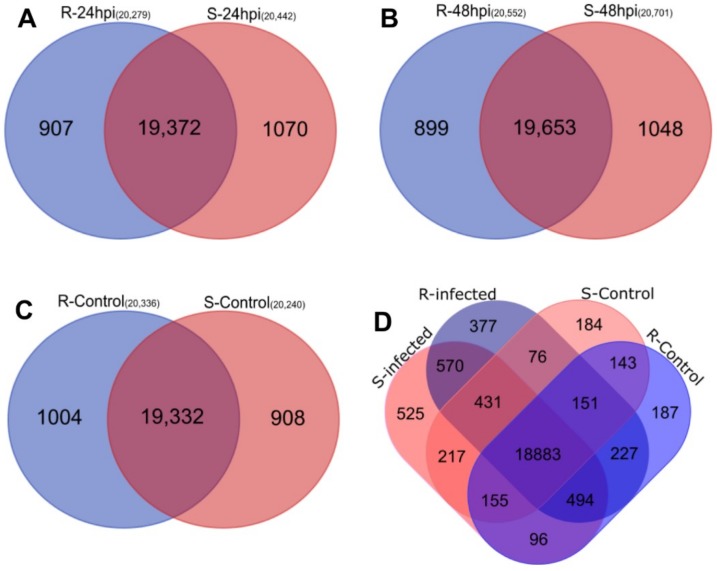
Venn diagrams presenting the overlap of expressed genes between resistant (Sp-R) and susceptible (Sp-S) wild tomato accessions. Pink circle represents Sp-R and blue is for Sp-S samples. (**A**) Comparison of *P. parasitica* infected Sp-R and Sp-S samples at 24 hpi. (**B**) Comparison of *P. parasitica* infected Sp-R and Sp-S samples at 48 hpi. (**C**) Comparison of mock inoculated R and S samples. (**D**) Combined comparison between all (mock and infected) Sp-R and Sp-S libraries.

**Figure 3 ijms-19-03735-f003:**
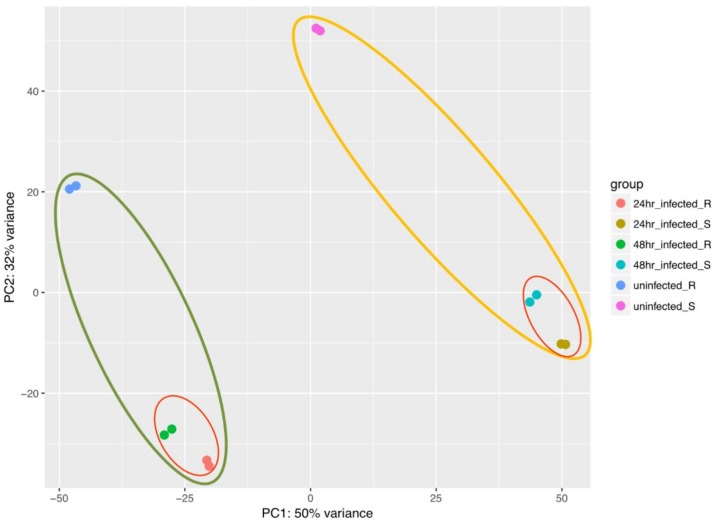
Principle component analysis (PCA) plot. Sp-R and Sp-S genotype factor showed variation among all their treated mock samples. Green oval is encompassing all Sp-R samples and red oval within the green oval is showing infected Sp-R samples at each time point. Yellow oval has enclosed all Sp-S samples within that red is encircling the infected ones. Color key presenting the sample groups is given at the left. Each colored dot represents a single sample. Replications are indicated by same color dots.

**Figure 4 ijms-19-03735-f004:**
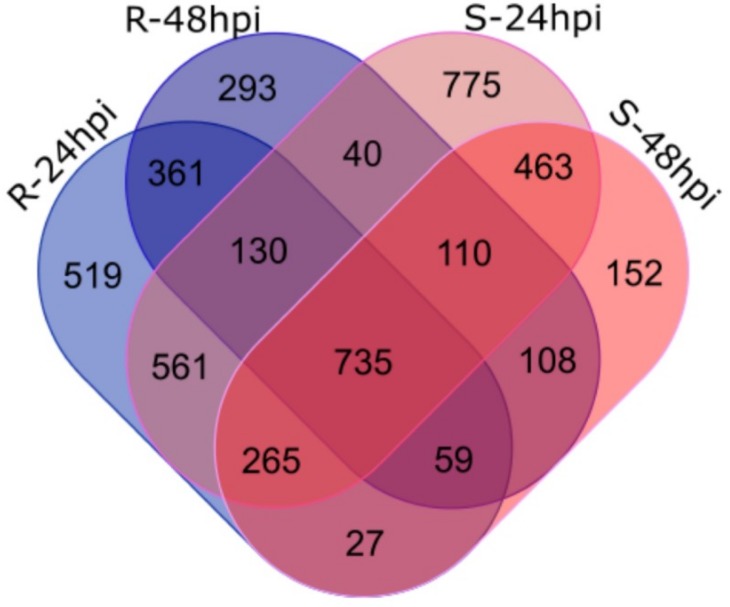
Venn diagram analysis of differentially expressed genes (DEGs). DEGs from treatment vs control comparisons of both Sp-R and Sp-S at 24 and 48 hpi. Pink ovals represent Sp-S treatment vs control and blue represents Sp-R treatment vs. control comparisons.

**Figure 5 ijms-19-03735-f005:**
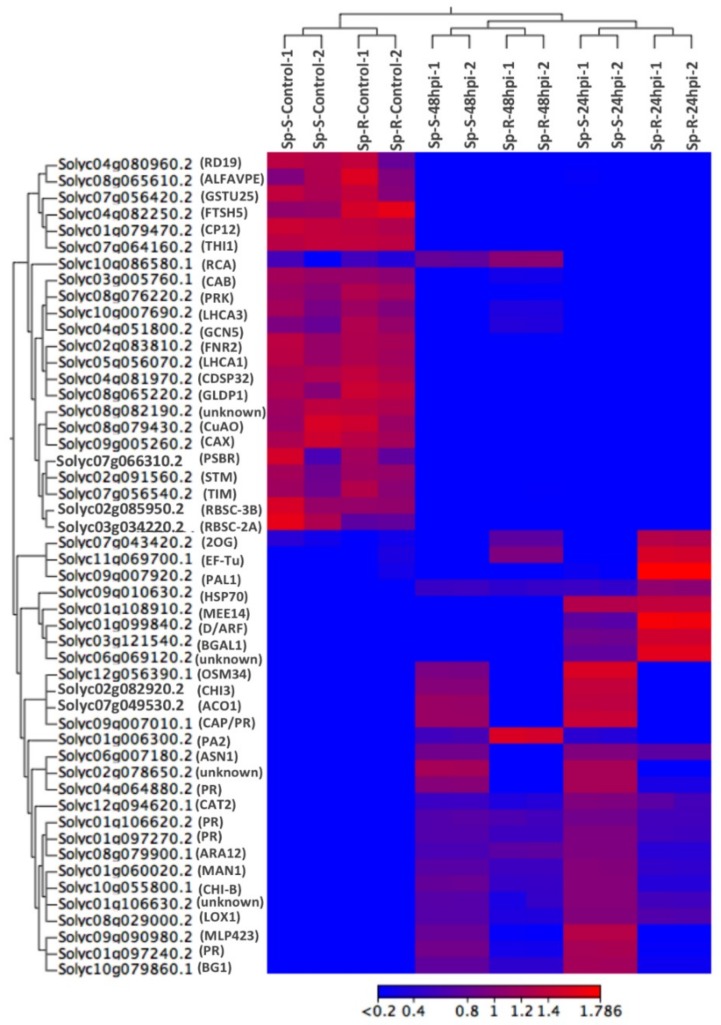
Heat map showing hierarchical cluster analysis of top 50 highly expressed genes across all samples. It shows that many core plant defense genes have differential expression profiles both among treatment vs. control as well as Sp-R vs. Sp-S conditions. Gradient scale is representing expression levels with red showing highest expression to blue with lowest expression.

**Figure 6 ijms-19-03735-f006:**
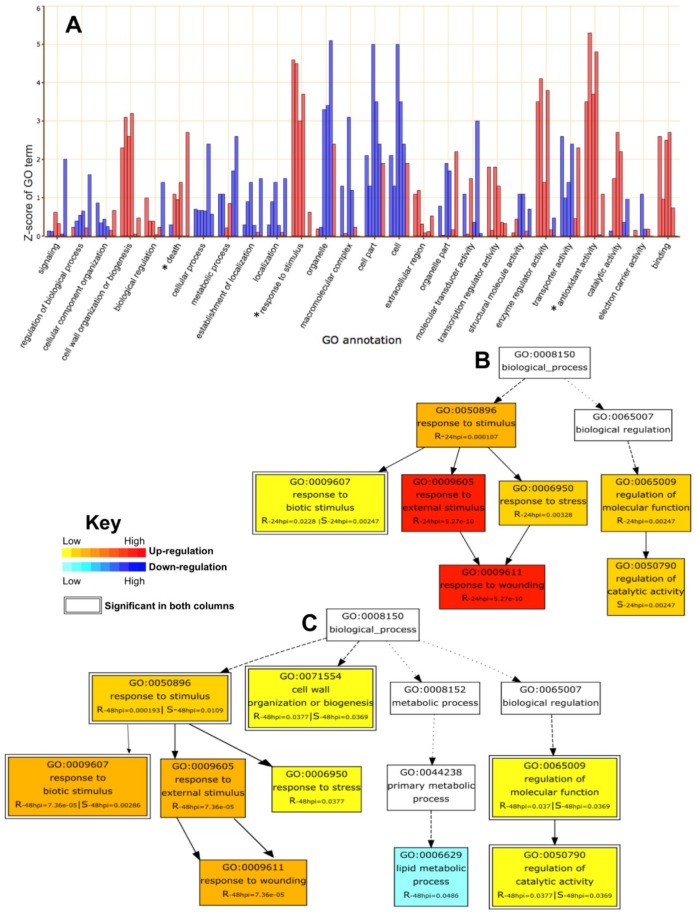
Gene set enrichment analysis by parametric analysis of gene set enrichment (PAGE) tool of AgriGO. (**A**). Bar chart representing GO annotations across all groups. Z-scores plotted on Y axis were calculated using fold change values of DEGs. Six bars are representing six differential expression comparisons among the samples. Starting from the left, first and second bars represent Sp-R-treated vs. Sp-R-control comparison at 24 and 48 hpi respectively, third and fourth bars are depicting Sp-S-treated vs. Sp-S-control at 24 and 48 hpi respectively and fifth and sixth bars are for Sp-R-treated vs. Sp-S-treated at 24 and 48 hpi respectively. Negative Z-score depicts down regulation, represented by blue bars whereas positive Z-score means upregulation, shown as red bars. Star indicates plant defense related annotations. (**B,C**) Comparative (Resistant vs. susceptible) GO biological process hierarchal analysis based on gene set enrichment analysis at 24 hpi (**B**) and 48 hpi (**C**) Significantly enriched upregulated terms are shown in color gradient red to yellow, red means highest level of upregulation and yellow means slightly upregulated whereas terms in white color box are not significantly enriched.Downregulation is showed in blue color gradient. Boxes that have double borders means that gene set is significantly enriched in both R and S samples.

**Figure 7 ijms-19-03735-f007:**
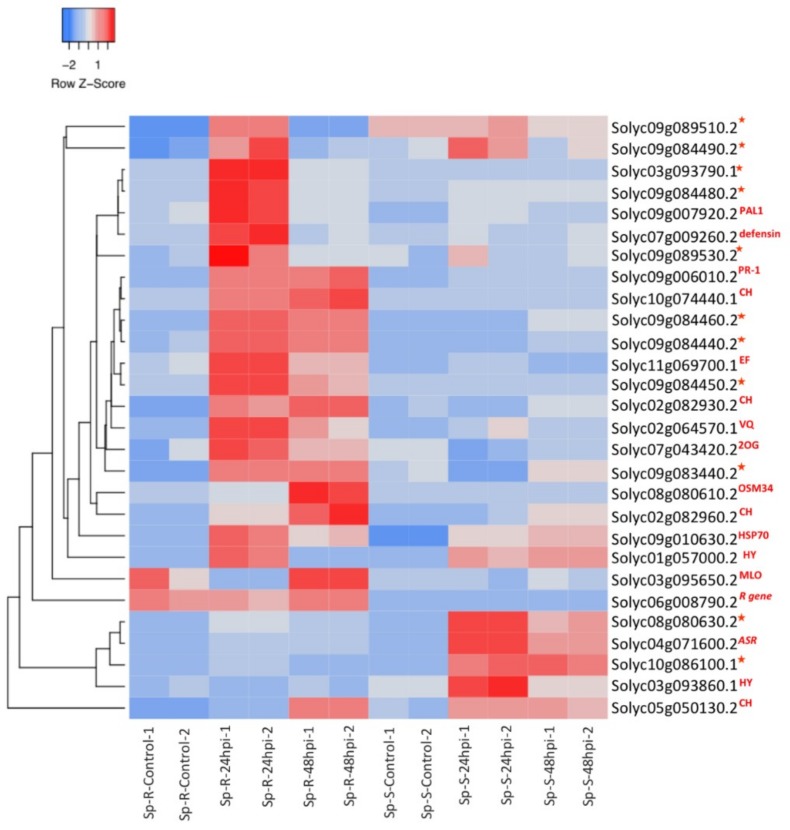
Heatmap showing expression profiles of core defense genes that could be putatively involved in resistance development against P. parasitica in Sp-R. Gradient scale is showing Z-scores of DEGs where red represents most induced expression and blue depicts highest repression. Gene identifiers with red stars on the right side are protease inhibitors. Other abbreviations used are: CH- chitinases, HY-hydrolases, VQ-VQ motif containing gene, 2OG-2OG-Fe oxygenase, EF-GTP binding Elongation factor Tu, OSM34-osmotin34, HSP70-heatshock protein 70, ASR- Abscisic acid stress ripening 5, PAL1-PHE ammonia lyase 1, and R gene-NB-ARC domain-containing disease resistance gene.

**Figure 8 ijms-19-03735-f008:**
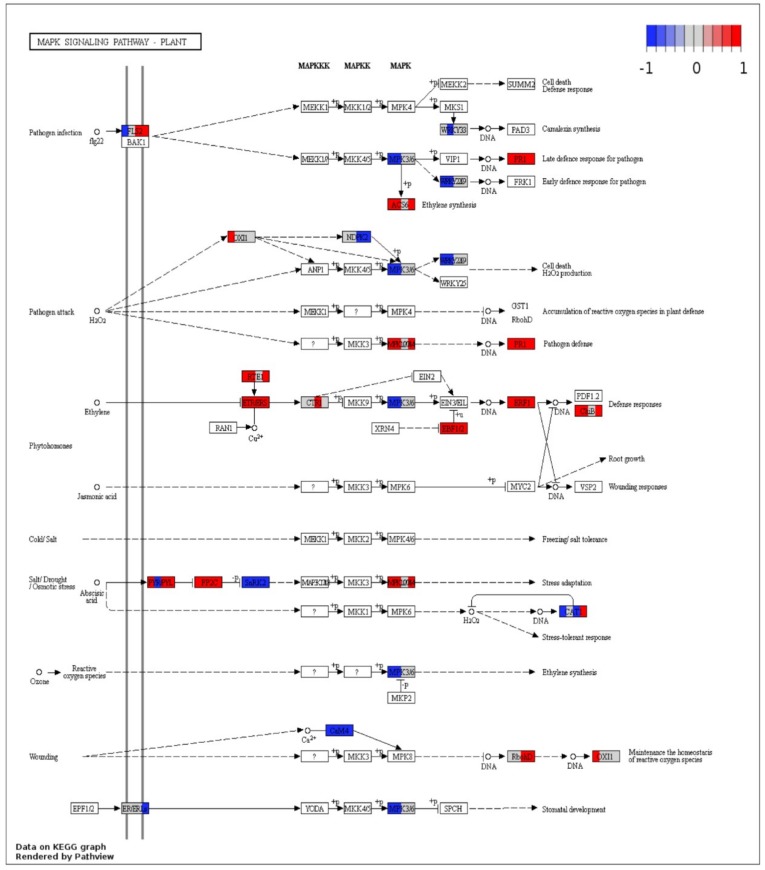
Visualization of DEGs involved in the mitogen activated protein kinase (MAPK) signaling pathway. Each box depicts DEGs from four treatments vs. control comparisons thus divided into four segments, the left half represents Sp-R treatment vs. control with first segment (left) 24 hpi and second 48 hpi. The right half is depicts Sp-S treatment vs. control with first segment (left) 24 hpi and second (right) 48 hpi. White box or no color fill means no DEG was assigned to that KO term. Color gradient represents log2 fold ratios with red representing upregulation and blue representing downregulation in treatments vs. mock samples. White segment in between colors means that term was not differentially expressed in the respective comparison.

**Figure 9 ijms-19-03735-f009:**
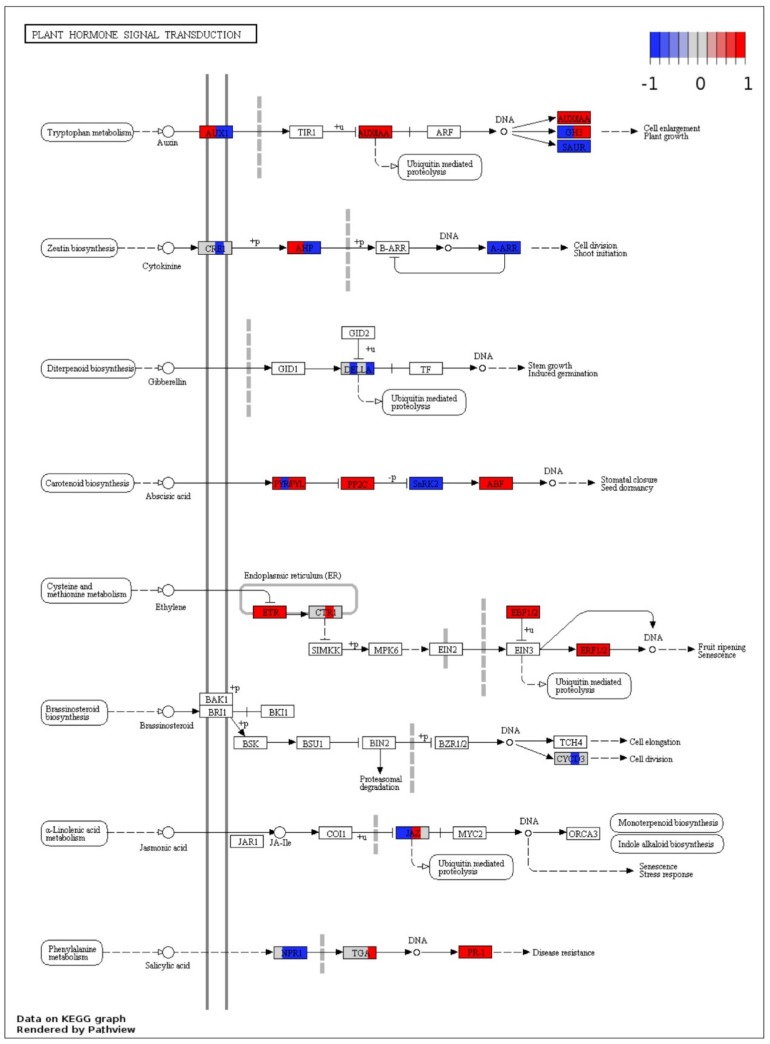
DEGs mapped on plant hormone signal transduction pathway. Each box depicts DEGs from four treatments vs. control comparisons thus divided into four segments, the left half represents Sp-R treatment vs. control with first segment (left) 24 hpi and second 48 hpi. The right half depicts Sp-S treatment vs. control with first segment (left) 24 hpi and second (right) 48 hpi. White box or no color fill means no DEG was assigned to that KO term. Color gradient represents log2 fold ratios with red representing upregulation and blue representing downregulation in treatments vs. mock samples. White segment in between colors means that term was not differentially expressed in the respective comparison.

**Figure 10 ijms-19-03735-f010:**
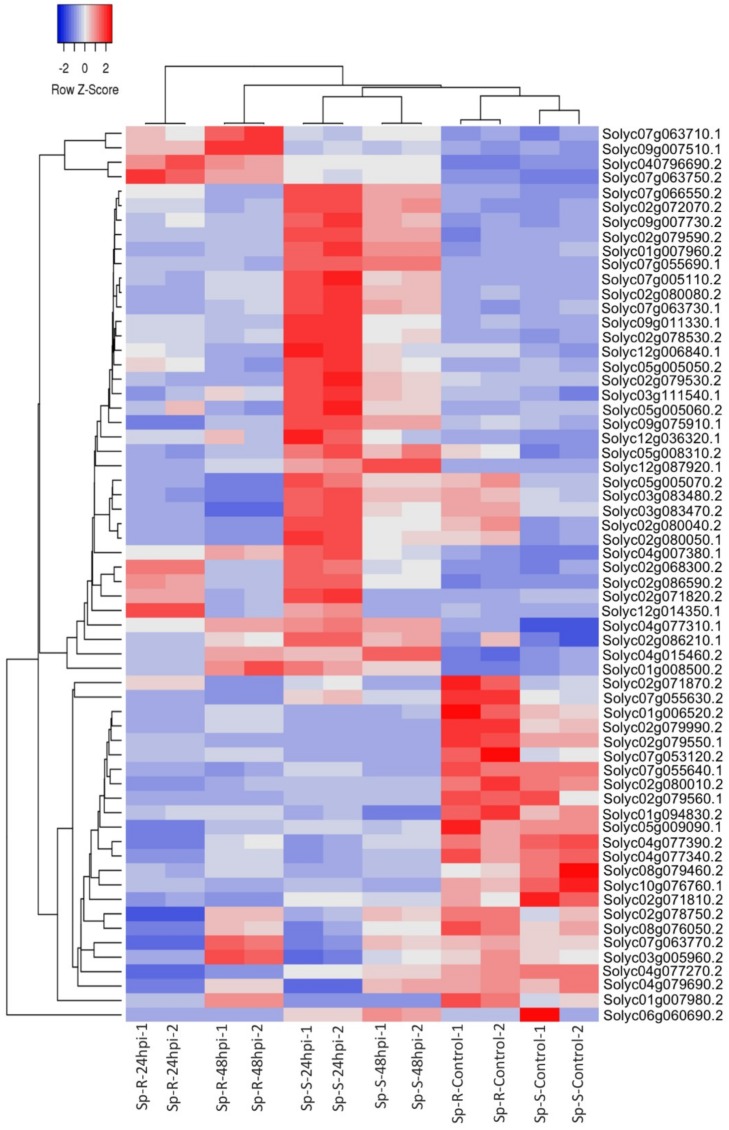
Heatmap showing expression profiles of DEGs encoding DUF26 type receptor like kinases (RLKs). The gradient scale shows scores of DEGs, where red represents most induced expression and blue depicts highest repression.

**Figure 11 ijms-19-03735-f011:**
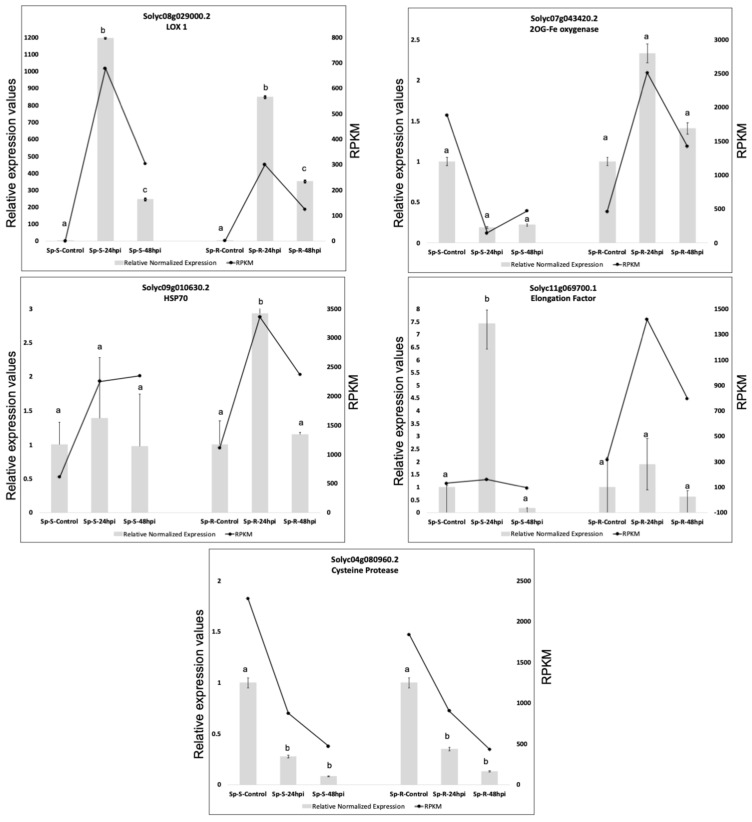
qRT-PCR based validation of plant defense related DEGs of wild tomato in response to *P. parasitica*. Expression levels of tested genes were normalized based on transcript levels of Actin gene. RPKM values calculated from RNA-seq are compared to relative expression values determined by qRT-PCR analysis. Relative expression values of infected samples were determined by using the average expression value of all replicates of a particular group. Standard deviation among replicates is represented by error bars. Different alphabets represent significant difference at *p*-value < 0.05.

**Table 1 ijms-19-03735-t001:** Summary statistics of RNA-seq data and mapping results.

Group	Sample Name	Raw Reads	Clean Reads	Mapped Reads (%)	Uniquely Mapped (%)	No: of Mapped Genes
1	R-mock (Rep 1)	27,886,208	23,047,994	97.93	81.06	21,436
	R-mock (Rep 2)	29,938,504	24,872,121	97.93	77.35	21,374
2	R-24 hpi (Rep 1)	25,731,844	21,627,411	97.97	85.44	21,222
	R-24 hpi (Rep 2)	27,210,026	22,621,023	97.96	79.42	21,175
3	R-48 hpi (Rep 1)	27,175,390	22,690,068	97.67	76.36	21,356
	R-48 hpi (Rep 2)	29,441,742	24,775,792	97.82	80.97	21,533
4	S-mock (Rep 1)	19,181,092	16,133,709	97.59	76.95	20,966
	S-mock (Rep 2)	27,938,220	23,227,317	97.07	84.29	21,381
5	S-24 hpi (Rep 1)	30,978,162	26,197,946	97.31	79.66	21,429
	S-24 hpi (Rep 2)	28,387,338	23,713,599	97.53	79.46	21,280
6	S-48 hpi (Rep 1)	28,563,832	23,854,528	96.27	76.36	21,631
	S-48 hpi (Rep 2)	26,601,354	21,881,482	97.36	77.45	21,565
	Total	329,033,712	274,642,990			

Group 1, 2 and 3 comprise of mock, 24 and 48 hpi *P. parasitica* treated samples of resistant accession (Sp-R) whereas 4, 5 and 6 represent mock, 24 and 48 hpi *P. parasitica* treated samples of susceptible accession (Sp-S) respectively.

**Table 2 ijms-19-03735-t002:** Comparison of Gene Ontology (GO) term enrichment analysis (by SEACOMPARE) of DEGs found in *P. parasiticat* infected vs. control samples of both resistant (Sp-R) and susceptible (Sp-S) wild tomato genotypes.

GO Term	Description	Color Model		No: of DEGs	
		R	S	R	S
GO:0003824	catalytic activity			899	974
GO:0004866	endopeptidase inhibitor activity			19	---
GO:0030414	peptidase inhibitor activity			20	---
GO:0019825	oxygen binding			52	57
GO:0016740	transferase activity			337	344
GO:0016798	hydrolase activity, acting on glycosyl bonds			47	53
GO:0016491	oxidoreductase activity			188	214
GO:0004364	glutathione transferase activity			---	18
GO:0004553	hydrolase activity, hydrolyzing O-glycosyl compounds			35	42
GO:0016758	transferase activity, transferring hexosyl groups			75	64
GO:0046527	glucosyltransferase activity			49	41
GO:0016757	transferase activity, transferring glycosyl groups			91	82
GO:0016168	chlorophyll binding			15	16
GO:0008194	UDP-glycosyltransferase activity			45	40
GO:0046906	tetrapyrrole binding			21	24
GO:0035251	UDP-glucosyltransferase activity			34	32
GO:0016705	oxidoreductase activity, acting on paired donors, with incorporation or reduction of molecular oxygen			46	45
GO:0004091	carboxylesterase activity			62	---
GO:0016759	cellulose synthase activity			15	---
GO:0016614	oxidoreductase activity, acting on CH-OH group of donors			37	41
GO:0015103	inorganic anion transmembrane transporter activity			22	25
GO:0005372	water transmembrane transporter activity			14	15
GO:0015250	water channel activity			14	15
GO:0022891	substrate-specific transmembrane transporter activity			---	127
GO:0022892	substrate-specific transporter activity			---	149
GO:0005215	transporter activity			---	178
GO:0022857	transmembrane transporter activity			---	147
GO:0022804	active transmembrane transporter activity			---	86
GO:0015291	secondary active transmembrane transporter activity			---	51
GO:0019203	carbohydrate phosphatase activity			---	11
GO:0008509	anion transmembrane transporter activity			---	34
GO:0015171	amino acid transmembrane transporter activity			---	19
GO:0016616	oxidoreductase activity, acting on the CH-OH group of donors, NAD or NADP as acceptor			---	34
GO:0005275	amine transmembrane transporter activity			---	20

Significance levels are based on enrichment and lowest FDR values with a cutoff of <0.05 that is indicated by color model mapping significance on a red to yellow gradient scale. Highly significant terms indicated by red. --- means no term assigned in that particular category.

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
