# Peer review of "Comparative Transcriptome Analysis between a Resistant and a Susceptible Wild Tomato Accession in Response to Phytophthora parasitica"

_ijms, 2018, doi:10.3390/ijms19123735_

Round 1
Reviewer 1 Report
The manuscript reviewed presents the results of comparative transcriptome analysis of two genotypes of wild tomato species Solanum pimpinellifolium which were contrast by the resistance to disease caused by Phytophtora parasitica. To reveal genes those transcriptional reprogramming was triggered by P. parasitica in the resistant and susceptible accessions the authors investigated their transcriptome profiles after inoculation with ionized water or zoospores of highly infectious P. parasitica strain. Among the genes the expression of which was specifically related to resistance in the resistant accession a number of pathogenesis related genes encoding for antimicrobial compounds were revealed. Moreover, for the resistant accession the repression of expression of several genes including powdery mildew susceptibility gene (Mlo) during resistance reaction was demonstrated.
The manuscript includes six sections – Abstract, Introduction, Results, Discussion, Materials and Methods, and References. The experimental data are illustrated with two tables, eleven informative figures, and supplementary files. The reference list contains 49 references.
The article undoubtedly represents new interesting data important for understanding genetic mechanisms controlling resistance of wild tomato spacies against P. parasitica.
Comments and suggestions.
1. The plant material should be described more completely (the origin, collection etc.). It is unclear whether of four known resistance genes (Ph-1, Ph-2, Ph-3 or Ph-5) was present in the resistant accession or its genotype was unknown.
2. Although principle component analysis very clearly illustrate differentiation between resistant and susceptible accessions very clearly (Figure 3) the conclusion on the role of genotype factor should be made with caution because of:
a limited number (two) of contrast accessions in the study
a limited number of treatments. In this connection it is necessary to explain why only the time points (24 and 48 hours) have been chosen for the treatments.
3. The text needs to be edited. It is necessary to decipher some abbreviations (for example, RPKM, MAPK) and to correct misprints (for example DET instead of DEG on lines 361, 556, 558).
Author Response
Thank you so much for positive feedback and valuable suggestions for improvement of the manuscripts. Given below is the point by point response to each comment/suggestion.
1. The plant material should be described more completely (the origin, collection etc.). It is unclear whether of four known resistance genes (Ph-1, Ph-2, Ph-3 or Ph-5) was present in the resistant accession or its genotype was unknown.
Homologs of these genes are present in the resistant accession but none of them is responsive to P. parasitica.
2. Although principle component analysis very clearly illustrate differentiation between resistant and susceptible accessions very clearly (Figure 3) the conclusion on the role of genotype factor should be made with caution because of:
a limited number (two) of contrast accessions in the study
a limited number of treatments. In this connection it is necessary to explain why only the time points (24 and 48 hours) have been chosen for the treatments.
We appreciate the reviewers concerns and we have tried to not overemphasize the role of the genotype factor as per suggestion. These two time points were selected to cover both the biotrophic phase (approximately 24 hrs after inoculation) as well as necrotrophic phase (48 hrs. after inoculation). These points have been explained in lines 113-118.
3. The text needs to be edited. It is necessary to decipher some abbreviations (for example, RPKM, MAPK) and to correct misprints (for example DET instead of DEG on lines 361, 556, 558).
Abbreviations are deciphered, and all corrections are done as per suggestions.
Reviewer 2 Report
Manuscript IJMS-39124 entitled “Comparative transcriptome analysis between a resistant and a susceptible wild tomato accession in response to Phytophthora parasitica” by Naveed and Ali describes a comparative study of the response of a fungal pathogen of two tomato cultivars, one resistant and other sensitive. Comparison of the defense response of these cultivars is performed at the transcriptome level (obtained with RNAseq technology). As a result of this comparison the authors are able to isolate several sets of genes which could be a good starting point of future studies in order to understand the molecular basis of the resistance of tomato to this pathogen.
The work is correct in terms of experimental design, data collection and transcriptome data processing. The only criticism that could be pointed out is the use of only two biological replicates for each condition, but the authors have demonstrated in a proper way the reproducibility of the replicates.
From my point of view is a well-written and properly discussed work. However, I have detected some minor issues that need to be attended prior to this final acceptation:
Lines 28-29: please remove this lines (keywords are included in lines 30-31)
Line 51: please consider to change the sentence into a something like “Through a molecular machinery coded by resistance genes” since genes are not physical structures that participate directly in any cell response.
Figure 1: Y axis legend should be detailed.
Table 1: Details of the samples should be included in table legend.
Line 162: a description of the results shown in the figure should not be present in the figure legend, since it is described in the text.
Figure 6: Please check color descriptions in the legend. Green color is absent in the figure.
Figure 11:
-please correct “QRT-PCR” with “qRT-PCR”
- Although the expression changes are evident, a statistical analysis should be performed to indicate which changes are significant.
-qRT-PCR data comes again from RNA of the same two biological replicates or a third or more biological replicates have been added to each condition?
-y-axis legend should be included
Line 421: please correct “gens” with “genes”
Line 446: Please check reference format of Osorio-herm 2016
Line 518: Methodology used to quantify the extent of the infection (data from figure 1c) should be detailed in “P parasitica inoculation” section
Line 565: Please correct reference list, numerical tags should be removed since the references are cited in alphabetical order.
Supplemental material: please check English in the names of the files, as well as in the table and figure legends. Some typos are present
Author Response
Thank you so much for positive feedback and valuable suggestions for improvement of the manuscripts. Given below is the point by point response to each comment/suggestion.
Lines 28-29: please remove these lines (keywords are included in lines 30-31)
Lines 28-29 are removed.
Line 51: please consider to change the sentence into a something like “Through a molecular machinery coded by resistance genes” since genes are not physical structures that participate directly in any cell response.
The sentence is being changed as per suggestion.
Figure 1: Y axis legend should be detailed.
Legend is added.
Table 1: Details of the samples should be included in table legend.
Table legend is modified as suggested.
Line 162: a description of the results shown in the figure should not be present in the figure legend, since it is described in the text.
Result description is removed.
Figure 6: Please check color descriptions in the legend. Green color is absent in the figure.
Figure legend is corrected. Red bars are depicting upregulation mistakenly written as green.
Figure 11:
-please correct “QRT-PCR” with “qRT-PCR”
Corrected
- Although the expression changes are evident, a statistical analysis should be performed to indicate which changes are significant.
Statistical analysis is done. Figure 11 is modified accordingly.
-qRT-PCR data comes again from RNA of the same two biological replicates or a third or more biological replicates have been added to each condition?
Same two biological replicates, each one in four technical replicates were used for qRT-PCR.
-y-axis legend should be included
Legends are added.
Line 421: please correct “gens” with “genes”
Corrected
Line 446: Please check reference format of Osorio-herm 2016
Reference is corrected.
Line 518: Methodology used to quantify the extent of the infection (data from figure 1c) should be detailed in “P parasitica inoculation” section
Details added.
Line 565: Please correct reference list, numerical tags should be removed since the references are cited in alphabetical order.
Removed
Supplemental material: please check English in the names of the files, as well as in the table and figure legends. Some typos are present
Corrected